# Forecasting Diabetic Complications from Brazilian Billing Codes with Time-Aware Attention

## Abstract

Predicting severe diabetic complications from longitudinal patient traces can enable proactive care. Where multi-institutional EHR integration is impractical, standardized health-insurance claims offer broad, longitudinal coverage despite clinical sparsity. We study this setting in Brazil's TUSS billing-code ecosystem and present a claims-only framework for forecasting complications (angiopathies, amputations, renal failure) 6–12 months ahead. TUSS codes are represented with skip-gram embeddings, and absolute timing is injected via fixed sinusoidal time embeddings added directly to event vectors; a BiLSTM with self-attention summarizes long, irregular histories. On anonymized data from 3.9 million individuals, the model achieves an AUC of 0.907 and an Average Precision of 0.631, outperforming capacity-matched baselines. Ablations show that temporal encoding and attention are complementary, with large gains only when combined. We further observe robust transfer to a second operator and concordant blinded field validations that surfaced previously unrecognized high-risk patients. While our contribution is a methodological instantiation rather than an architectural novelty, the work offers a careful case study of claims-only prediction at a national scale, design lessons for modeling sparse transactional health data, and practical evidence for its utility in real-world risk stratification.

## 1 Introduction

Diabetes mellitus is a major global health challenge, driving substantial morbidity, mortality, and costs through late-stage complications such as angiopathies, amputations, and renal failure (22). Early identification of high-risk patients could enable preventive interventions, yet building accurate predictors typically requires large, longitudinal cohorts with consistent follow-up. Electronic Health Records (EHRs) provide rich clinical detail, but assembling multi-institutional corpora suitable for robust learning is hindered by interoperability barriers, heterogeneous semantics, and privacy constraints (26). As a result, many predictive studies remain confined to single centers or narrowly scoped populations.

Administrative claims offer a complementary substrate: they are standardized for reimbursement and cover large populations over long horizons, albeit with sparser clinical detail. In Brazil's supplementary health system, the *Padrão TISS* standard governs information exchange between payers and providers, and the *TUSS* terminology unifies codes for procedures, medications, and materials across the private sector (18). This creates a nationally consistent vocabulary of transactional events that parallels international billing standards such as ICD and CPT (11). While claims are not designed to capture full clinical intent, their scale and regularity make them attractive for risk prediction when EHR integration is impractical.

This work examines whether claims-only longitudinal histories, encoded with TUSS codes, can support early prediction of severe diabetic complications. Focusing on a single clinical area and data source, the study provides a large-scale case analysis of the opportunities and limitations of claims-based modeling in a national ecosystem. Three challenges are central: (i) representing a large and evolving code vocabulary, (ii) incorporating absolute time information from irregular event

streams, and (iii) coping with the sparsity and noise inherent to transactional data, which reflect reimbursement practices as well as clinical state.

To address these, we combine dense TUSS embeddings, absolute sinusoidal *Time Embeddings* (23), and a BiLSTM with self-attention to summarize patient histories, augmented with simple demographic covariates. While modern Transformers dominate recent clinical sequence modeling (21), this study investigates how far a parameter-efficient recurrent backbone can be pushed when paired with principled temporal and code representations.

**Contributions.** This paper makes the following contributions:

1. **National-scale claims case study.** The first large-scale analysis of diabetic complication prediction using Brazil's TUSS-coded claims, with transparent documentation of cohort construction and modeling design.

2. **Temporal design insights.** Evidence that absolute sinusoidal *Time Embeddings* and self-attention are complementary: each adds modest benefit alone, but their combination yields the largest gains in AUC and average precision.

3. **Reproducibility and fairness.** Clear reporting of sampling strategies, ablations, and evaluation protocols (ROC/PR curves), alongside discussion of claims-specific biases, generalizability, and fairness considerations.

On anonymized longitudinal data from ∼3.9 million individuals across two health operators, the proposed pipeline predicts severe diabetic complications 6–12 months in advance with strong performance, outperforming capacity-matched baselines. Beyond headline metrics, the analysis provides practical lessons for modeling sparse transactional health data at national scale.

## 2 RELATED WORK

Deep sequence models have become central to modeling longitudinal healthcare data for risk prediction, diagnosis forecasting, and patient stratification. Early architectures based on recurrent networks and attention mechanisms, such as RETAIN, established strong baselines for visit-level prediction tasks, while more recent approaches increasingly leverage self-attention to capture long-range dependencies and heterogeneous inputs (8; 17). Transformer-based models specialized for EHR, including BEHRT and its successors, incorporate visit position and patient age directly into embeddings and consistently outperform recurrent networks on multi-label diagnosis prediction; recent surveys document their rapid adoption across both structured and unstructured modalities (16; 21; 10). Temporal convolutional networks (TCNs) have also emerged as competitive alternatives for handling long clinical sequences (3; 24).

Representation learning for medical codes is another active area. One-hot encodings of diagnosis, procedure, or medication codes are high-dimensional and fail to capture similarity across concepts. Learned embeddings, adapted from natural language processing, map codes to dense vectors using co-occurrence statistics and have been shown to improve downstream prediction tasks (19; 8; 6). Beyond visit-sequence embeddings such as Med2Vec, more recent approaches draw on large multimodal corpora or code descriptions processed with language models, enabling alignment across ontologies such as ICD, CPT, NDC, and SNOMED (4; 6).

Handling irregular sampling remains a key methodological challenge in clinical time series. Architectures such as GRU-D, T-LSTM, and Phased-LSTM introduce explicit modeling of elapsed time or decay to address missingness and non-uniform observation intervals (7; 5; 20). More recently, self-supervised Transformers for irregular series and continuous-time neural controlled differential equations (CDEs) aim to mitigate sparsity without requiring resampling (25; 14). Complementary to architectural innovation, several approaches inject explicit time embeddings into inputs. Time2Vec learns periodic and time-aware embeddings, while continuous-time positional encodings and sinusoidal date embeddings provide alternative formulations for irregular sequences (13; 15; 23).

Administrative claims data represent a distinct modeling context. Unlike EHRs, claims are collected primarily for reimbursement, resulting in broad longitudinal coverage but comparatively sparse clinical detail. In health systems where ICD diagnosis codes and CPT/HCPCS procedure codes

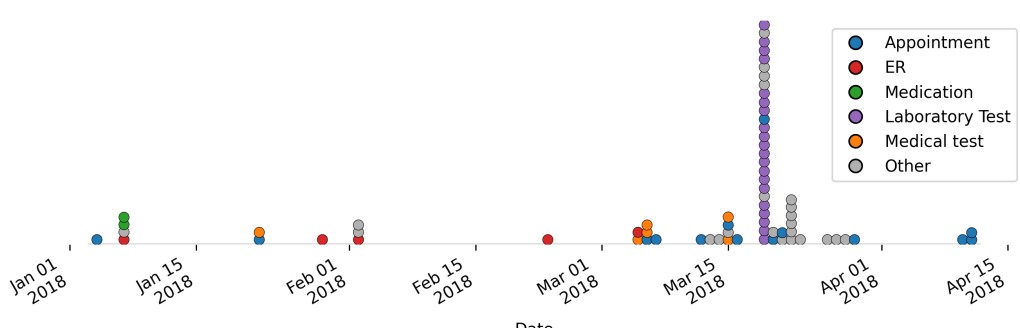

Figure 1: Example of a TUSS claim data series of a patient.

are included, claims datasets have supported robust disease prediction and population-level risk stratification (9; 12; 1). In Brazil's supplementary health system, however, the TISS standard mandates the use of TUSS codes for procedures, materials, medications, and daily rates (2). These codes are primarily transactional and often lack standardized diagnosis fields, making the prediction task more reliant on utilization patterns. This contrasts with claims datasets in other regions and raises specific challenges for embedding design and interoperability, including potential mappings to international terminologies such as LOINC and SNOMED (see Table 2).

Recent discussions of fairness in health AI emphasize that claims-derived models are shaped by provider incentives and policy structures, which may propagate inequities. Best practices now recommend explicit consideration of subgroup performance, bias sources, and mitigation strategies when deploying predictive systems in sensitive clinical domains.

Table 1: TUSS Examples

| TUSS | Description |
|---|---|
| 10101012 | Clinic Visit |
| 10101039 | Emergency Room Care |
| 40901475 | Color Doppler of Aorta and Iliac Arteries |
| 90019415 | Dipyrone 500mg |

Table 2: TUSS Interoperability

| Terminology | Hemoglobin A1c |
|---|---|
| TUSS | 40302733 |
| CPT | 83037 |
| LOINC | 4548-4 |
| SNOMED CT | 43396009 |

## 3 METHODOLOGY

### 3.1 DATA AND COHORT DEFINITION

We analyze anonymized longitudinal claims from two Brazilian health insurers that are both compliant with the national TISS/TUSS standard. **Operator 1** (training/evaluation) includes ∼3.9M beneficiaries and 62.7B claim lines (Jan/2013–Sep/2020). **Operator 2** (transfer validation) covers ∼628,779 beneficiaries (2006–2019) with typically longer individual histories. Each record is a tuple (TUSS code, service date), with demographics (age, sex). In Figure 1, there is a real example of a TUSS series of a patient from Operator 1.

Because diagnosis fields are inconsistent, a proxy diabetes cohort is defined via repeated HbA1c testing: individuals with ≥ 2 HbA1c exams (TUSS 40302733) within 12 months are included. This yields ∼105k patients in Operator 1 and ∼33k in Operator 2.

We forecast the first severe diabetes-related complication 6–12 months ahead, using curated TUSS sets for angiopathy, amputation, and renal failure (Appendix A). In Operator 1, 1,019 patients experience an outcome. For each patient, we extract all (code, date) events in a 90–720 day observation window before an index date. Positive sequences end ≥180 days before the first complication; negatives come from patients without complications. No information from the prediction horizon leaks into the observation window.

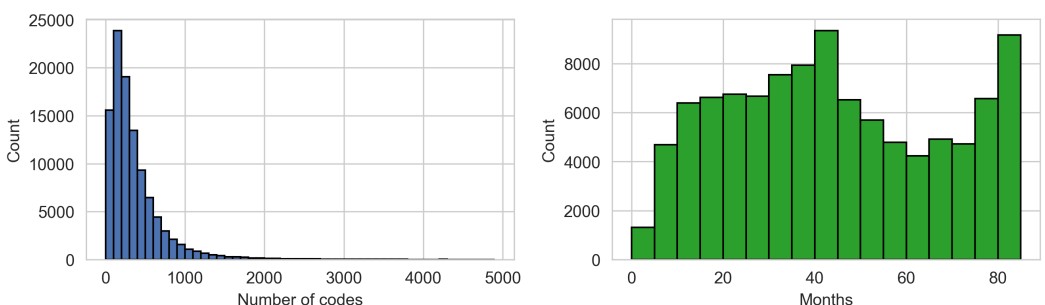

Figure 2: Data length distribution based on the number of individual TUSS codes and time length.

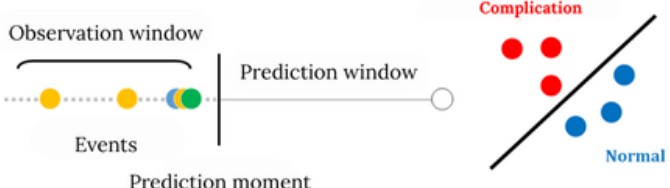

Figure 3: Modeling scheme where the prediction is based on the probability of a complication code occurring after more than six months.

The Operator 1 diabetes cohort used for supervised training includes 105k patients, with 57% female (60k) and 43% male (45k). The average age is 47 years (SD 17). Figure 2 shows the distribution of sequence lengths: most patients have fewer than 1,000 claim records, yet these records typically span multiple years, with some histories extending up to the maximum window of 92 months.

Validation/test preserves natural prevalence. During training, positives are oversampled within mini-batches to 1:1 to stabilize optimization under ~1% prevalence.

### 3.2 INPUT REPRESENTATION

We learn $d_{\text{TUSS}} = 200$-dimensional embeddings for ~170k unique TUSS codes via skip-gram with negative sampling over the whole Operator 1 dataset, treating each beneficiary history as a document (context window 15).

Absolute time is encoded with fixed sinusoidal embeddings $\mathbf{te}_t \in \mathbb{R}^{200}$:

$$\mathbf{te}_t[2i] = \sin\left(\tfrac{date_t}{P^{2i/d}}\right), \quad \mathbf{te}_t[2i+1] = \cos\left(\tfrac{date_t}{P^{2i/d}}\right),$$

with $P=10,000$ days. This provides continuous absolute time without modifying the backbone.

Each event vector is

$$\mathbf{e}_t = \mathbf{tuss}_t + \mathbf{te}_t \in \mathbb{R}^{200}.$$

Demographics are appended at the sequence level: sex (one-hot) and age (normalized, capped at 120).

### 3.3 MEDATTENTION ARCHITECTURE

The used architecture schematic can be seen in Figure 4. Given a sequence $E = (\mathbf{e}_1, \ldots, \mathbf{e}_L)$, $L \leq 500$:

1. **BiLSTM.** Hidden size 128 per direction, producing $H \in \mathbb{R}^{L \times 256}$.

2. **Self-attention pooling.** Following structured attention:

$$\mathbf{u} = \tanh(HW_1 + \mathbf{b}_1), \quad \boldsymbol{\alpha} = \text{softmax}(\mathbf{u}\mathbf{w}_2 + b_2), \quad \mathbf{c} = \boldsymbol{\alpha}^\top H.$$

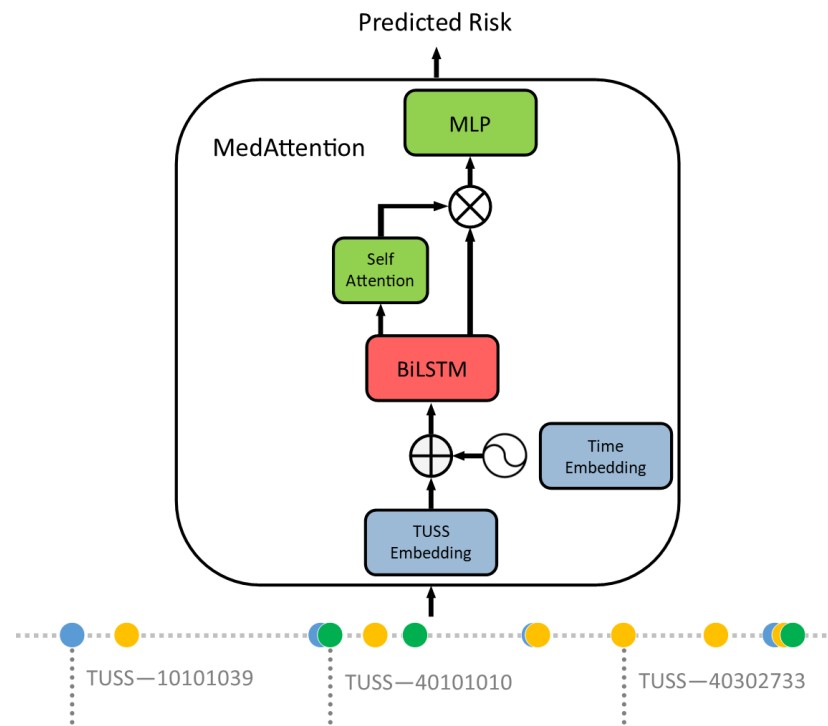

Figure 4: Proposed framework with all connected components.

3. **Classifier.** Concatenate **c** with age/sex and pass through a 2-layer MLP (64 units, ReLU, dropout 0.4) to output complication probability.

## 3.4 BASELINES AND CAPACITY CONTROL

We compare against: (i) a 4-layer MLP, (ii) a 6-layer, 8-head Transformer, and (iii) a 3-block Temporal Convolutional Network (TCN). All models use the same inputs, splits, optimizer, and training budget. Parameter counts are controlled for fairness: MedAttention $\sim$35M, Transformer $\sim$41M, TCN $\sim$35M, and MLP $\sim$60M. A BiLSTM backbone was retained for prospective validation feasibility; however, contemporary Transformer/TCN baselines are also included for completeness.

## 3.5 TRAINING AND REGULARIZATION

Models are implemented in PyTorch and trained with SGD (momentum 0.9, learning rate 0.01, $L_2=10^{-6}$). Regularization includes token-level dropout (0.2) and LockedDropout (0.2). Mini-batches (size 128) oversample positives to 1:1 during training. Loss is binary cross-entropy. Early stopping uses validation AUC.

## 3.6 EVALUATION PROTOCOL

Operator 1 is split at the patient level into train/validation/test (70/15/15), stratified by outcome. We report ROC AUC and Average Precision (AP) as primary threshold-independent metrics under imbalance, and $F_1$ at the validation-optimal threshold. Metrics are averaged over 10 runs (mean$\pm$sd). 95% CIs for AUC are obtained via bootstrap (Appendix C). For transfer, a model trained on Operator 1 is evaluated directly on Operator 2 (frozen encoder; threshold chosen on Operator 2 validation).

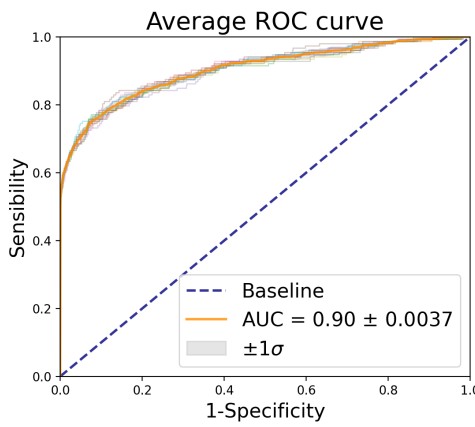 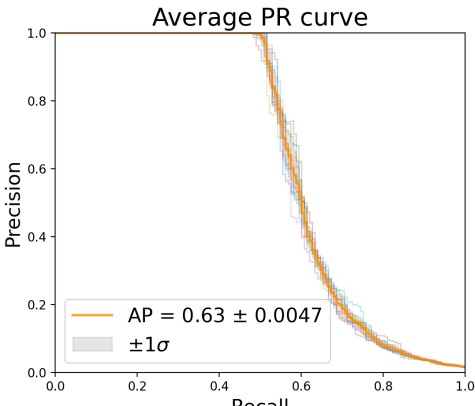

Figure 5: ROC curve of MedAttention          Figure 6: Precision-Recall curve of MedAttention

Table 3: Predictive performance on Operator 1 (mean $\pm$ sd over 10 runs). Best results in bold.

| Model | AUC | $F_1$ | AP |
|---|---|---|---|
| MLP | $0.781 \pm 0.013$ | $0.189 \pm 0.064$ | $0.234 \pm 0.035$ |
| TCN | $0.750 \pm 0.001$ | $0.064 \pm 0.002$ | $0.051 \pm 0.001$ |
| Transformer | $0.875 \pm 0.005$ | $0.279 \pm 0.049$ | $0.641 \pm 0.011$ |
| **MedAttention** | $0.907 \pm 0.003$ | $0.334 \pm 0.051$ | $0.631 \pm 0.003$ |

## 4 RESULTS

### 4.1 EVALUATION SETUP

Given severe class imbalance ($\sim$1% positives), we treat threshold-independent metrics as primary and report ROC AUC and Average Precision (AP). Thresholded $F_1$ scores are reported at validation-selected operating points. Full ROC and PR curves appear in Fig. 5–6.

### 4.2 COMPLICATION PREDICTION PERFORMANCE

Across 10 runs on Operator 1, **MedAttention achieves an AUC of** $0.907 \pm 0.003$ **and AP of** $0.631 \pm 0.003$, outperforming capacity-matched baselines (Table 3). The Transformer baseline reaches a slightly higher AP (0.641) but lower AUC (0.875). MedAttention also yields the best $F_1$ score ($0.334 \pm 0.051$), reflecting stronger thresholded discrimination. These results indicate that combining recurrent modeling with explicit absolute-time features and self-attention produces robust risk stratification from sparse claims sequences.

### 4.3 ABLATION STUDY

Ablations on Operator 1 (Table 4) show that neither Time Embeddings (TE) nor self-attention (Att) alone is sufficient: TE alone provides no benefit, and Att alone yields modest gains. Only their combination (BiLSTM+TE+Att, i.e., MedAttention) produces large improvements (AUC 0.907, AP 0.631). This supports the view that TE supplies temporally grounded signals which attention then exploits to emphasize clinically informative events.

### 4.4 TRANSFER AND FIELD VALIDATION

A model trained on Operator 1 generalized well to Operator 2 without retraining (AUC 0.92, AP 0.70), approaching the performance of a model trained natively on Operator 2 (AUC 0.95, AP 0.80). To assess real-world utility, both operators conducted blinded field validations in which beneficiaries with predicted complication risk above 80% were flagged for review. Follow-up confirmed that

Table 4: Ablation results on Operator 1. All models share the training setup.

| Model Configuration | AUC | $F_1$ | AP |
|---|---|---|---|
| BiLSTM | 0.741 | 0.065 | 0.050 |
| BiLSTM + TE | 0.735 | 0.064 | 0.047 |
| BiLSTM + Att | 0.817 | 0.083 | 0.089 |
| **BiLSTM + Att + TE (MedAttention)** | 0.907 | 0.334 | 0.631 |

these high-risk groups exhibited substantially higher rates of adverse outcomes (hospitalizations, ICU admissions, mortality) and markedly higher healthcare costs compared to the broader diabetic population. Notably, 41 of the 140 high-risk patients flagged in Operator 2 were not previously enrolled in monitoring programs but were subsequently reviewed by clinical experts and confirmed as high-risk diabetics, leading to their inclusion in targeted monitoring. This demonstrates the model's ability to surface previously unrecognized patients in need of proactive care.

At Operator 1, 243 beneficiaries were flagged as high-risk with predicted complication risk above 80% but no prior complication codes. Specialists examined outcomes to confirm patient status. Among the flagged cohort, two deaths related to chronic disease were recorded, along with multiple serious events including strokes (6), heart disease (31), renal failure (12), and peripheral vasculopathy (2). A substantial fraction experienced hospitalizations (34%) or ICU admissions (16%). Many were already under monitoring, but those not yet enrolled were incorporated into existing care-management programs.

Misclassifications were primarily linked to utilization patterns unrelated to diabetes but resembling high-acuity care, such as oncology treatments and high-risk pregnancies. These cases highlight an intrinsic limitation of claims-only modeling, where signals reflect utilization intensity rather than explicit clinical intent. Despite this, the blinded evaluations led both operators to validate the model as a useful tool for targeted monitoring and proactive care management.

### 4.5 RISK FACTOR ANALYSIS

Spearman correlations between per-patient code frequencies and predicted risk (Operator 2) reveal that ∼89% of codes correlate positively with risk, consistent with utilization intensity reflecting acuity. Most associations are weak (0–0.2), suggesting the model relies on combinations and timing of events rather than single markers. Positively associated categories include hospitalization/ICU rates and cardiovascular/renal markers (e.g., metoprolol, furosemide, morphine, creatinine, BNP). Negative associations arise from outpatient behavioral health codes and Type 1 diabetes immunoassays (anti-GAD). These patterns align with clinical expectations while underscoring that predictive signals in claims are compositional and temporal.

## 5 DISCUSSION

This study demonstrates that clinically actionable risk stratification is feasible using claims-only longitudinal histories encoded with the Brazilian TUSS standard. Although the backbone is deliberately simple, the combination of distributional embeddings for high-cardinality billing codes, absolute sinusoidal Time Embeddings (TE), and sequence-level self-attention yielded strong and consistent gains under extreme imbalance. Ablations confirm a clear synergy: TE alone provides little benefit, and attention alone adds modest discrimination, but together they produce large improvements in threshold-independent metrics.

External evaluation shows that a model trained on a large national operator transfers effectively to a distinct regional operator, despite differences in benefit mix and care pathways. This suggests that standardized coding and reimbursement processes induce transferable structure across claims populations. At the same time, performance portability should be viewed as a starting point: recalibration and monitoring will remain necessary as coding practices and policies evolve.

Overall, these findings provide design lessons for modeling sparse, irregular, transactional sequences. They clarify when a parameter-efficient recurrent backbone, augmented with principled time encoding and attention, can remain competitive against more complex architectures in claims-centric settings.

## 6 LIMITATIONS AND BROADER IMPACT

Administrative claims are collected for reimbursement rather than clinical intent. They omit nuance, conflate severity with billing intensity, and reflect provider incentives, which may introduce misclassification, spurious associations, and fairness risks. Our diabetes cohort definition, based on repeated HbA1c tests, excludes underdiagnosed or poorly monitored patients, limiting coverage. The study focuses on a single disease and two operators; although transfer results are promising, broader validation across conditions, payers, and time periods is required. Concept drift (e.g., policy updates, coding changes) further motivates ongoing monitoring.

Claims encode historical access and utilization patterns that differ across demographic and socioeconomic groups. Without safeguards, models may reproduce and amplify inequities, echoing well-documented failures of cost-based risk adjustment. Responsible use, therefore, requires: (i) clear intended-use statements, (ii) subgroup performance and calibration audits, (iii) fairness-aware recalibration where needed, (iv) human-in-the-loop oversight, and (v) transparent reporting following contemporary guidelines.

Beyond methodological insights, this work highlights the practical value of routinely collected claims data for population health management. In Brazil and comparable health systems, complications such as amputations and renal failure are leading drivers of disability and cost; early identification of high-risk individuals could enable targeted monitoring and preventive interventions that reduce avoidable hospitalizations, improve quality of life, and lower expenditures. Because billing standards such as ICD and CPT serve similar roles internationally, the design lessons drawn from TUSS may generalize to other claims ecosystems, broadening the applicability of claims-based AI beyond Brazil. More broadly, this study illustrates how scalable, non-invasive data sources can complement electronic health records when integration is infeasible, opening a pathway toward equitable, proactive care at a national scale.

## 7 CONCLUSION

We presented a claims-only framework for forecasting severe diabetic complications from Brazil's standardized TUSS billing data. By combining learned embeddings of billing codes with absolute Time Embeddings and a BiLSTM with self-attention, the proposed MedAttention model achieved strong discrimination on a large national cohort (AUC 0.907, AP 0.631) and transferred effectively to a second operator with different population and benefit mix. Ablations show that explicit absolute-time signals and attention are complementary, yielding the largest gains when combined.

The key practical takeaway is that routinely collected claims can support proactive risk stratification when multi-institutional EHR integration is infeasible. Responsible deployment requires careful operating-point selection, calibration monitoring, and governance to mitigate claims-specific biases. Looking ahead, promising directions include capacity-controlled comparisons with long-sequence Transformers and TCNs, richer interpretability of code–time interactions, fairness-aware recalibration and subgroup audits, and prospective impact evaluation in real care-management programs. We position these findings not as architectural novelty but as design lessons for modeling sparse claims sequences at the national scale.

## 8 REPRODUCIBILITY STATEMENT

All methodological details necessary to reproduce our results are included in the paper and appendices. We describe data preprocessing (cohort selection, outcome definitions, observation windows), model architecture (embedding sizes, time encoding, BiLSTM/attention design), baselines with parameter counts, training setup (optimizer, regularization, hyperparameters), and evaluation protocols (splits, metrics, calibration). The anonymized claims datasets analyzed here are not publicly shareable due to contractual and privacy restrictions, and we do not release source code. However, the framework is

fully specified such that researchers can implement it with comparable claims or EHR datasets in other health systems.

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

## A  COMPLICATION TARGET TUSS CODES

| TUSS Code | Description |
| --- | --- |
| 20103026 | Bilateral amputation stump preparation |
| 20103034 | Bilateral amputation (prosthetic training) |
| 20103042 | Unilateral amputation (stump preparation) |
| 20103050 | Unilateral amputation prosthetic training |
| 30718015 | Amputation at arm level |
| 30720036 | Amputation at forearm level - surgical treatment |
| 30721105 | Wrist and forearm amputation stump - revision |
| 30722063 | Amputation at metacarpal level surgical treatment |
| 30722071 | Finger amputation |
| 30722080 | Transmetacarpal amputation |
| 30722098 | Transmetacarpal amputation with finger transposition |
| 30722241 | Digital amputation stump revision |
| 30725038 | Amputation at thigh level surgical treatment |
| 30727049 | Leg amputation - surgical treatment |
| 30728010 | Amputation at ankle level surgical treatment |
| 30729017 | Amputation at foot level - surgical treatment |

| TUSS Code | Description |
| --- | --- |
| 30729025 | Amputation/disarticulation of toes per segment surgical treatment |
| 40601234 | Diagnostic procedure in limb amputation - non-oncological cause |
| 30724120 | Hip disarticulation surgical treatment |
| 30726069 | Knee disarticulation surgical treatment |
| 30101280 | Surgical debridement - per topographic unit (TU) |
| 30730031 | Surgical debridement of wounds or extremities |
| 30729122 | Fasciotomy or plantar fascia resection - surgical treatment |
| 30730074 | Fasciotomy |
| 30730082 | Fasciotomy - per compartment |
| 30730090 | Decompressive fasciotomies |
| 30906423 | Upper limb arterial revascularization |
| 30906113 | Transoperative transluminal angioplasty - per artery |
| 30912024 | Transluminal angioplasty of the aorta or branches or of the pulmonary artery and branches (per vessel) |
| 30912032 | Percutaneous transluminal angioplasty of multiple vessels, with stent implantation |
| 30912040 | Percutaneous transluminal balloon angioplasty 1 vessel |
| 40813070 | Supra-aortic trunk angioplasty |
| 40813177 | Percutaneous transluminal angioplasty |
| 40813185 | Percutaneous transluminal angioplasty for treatment of arterial obstruction |
| 30909031 | Chronic hemodialysis (per session) |
| 30909139 | Acute case hemodepuration |
| 30909023 | Continuous hemodialysis (12h) |
| 30909015 | Acute case hemodepuration |

