# OpenReview forum: "Forecasting Diabetic Complications from Brazilian Billing Codes with Time-Aware Attention"
_ICLR.cc/2026/Conference — Submitted to ICLR 2026_

### Official Review · Reviewer_Gysh · 2025-10-28

**Soundness:** 1
**Presentation:** 2
**Contribution:** 1
**Rating:** 0
**Confidence:** 5

**Summary:**

This paper proposes to improve diabetes complication prediction by combining BiLSTM with a time embedding and self attention.

**Strengths:**

- The large dataset evaluation is good, even with the issues with it being private
- AUROC/AP are very good metrics (not a fan of F1 for a variety of reasons, but the important stuff is there)

**Weaknesses:**

- Many missing baselines. In particular, the lack of a gradient boosted tree baseline is critical, as those are both highly performing and require minimal setup.
- I would also like to see a logistic regression model, mainly because the claimed MLP performance is so low.
- Lack of evaluation on a public dataset, ideally MIMIC-IV is a significant downside of this paper as it makes reproducibility more challenging
- Incorrect grammar in various parts of the paper "Validation/test preserves natural prevalence." doesn't make any sense. The writing in general needs to be improved.
- No discussion of hyperparameter tuning? Especially learning rate, which is very important for both the proposed method and baselines
- Absolute time as a feature should only be used when time splitting is also used for evaluation (where some years are held out for testing). Otherwise you risk severe absolute time overfitting problems. I don't think time splitting was used during evaluation?
- Unclearn novelty, especially because sinusoidal time embeddings are pretty common. See for example EHRSHOT. The related work section in general should be more comprehensive
- It would be good to evaluate on more tasks than just one, diabetes complications. I usually like to see a mix of acute and chronic predictions so we can better understand how a method performs in various environments.
- Static skip-gram embeddings are not ideal for this setup, with this many patients. You should allow the embeddings to be learned as part of the training (especially since you have so many patients).

**Questions:**

See weaknesses

---

### Official Review · Reviewer_Tkjq · 2025-10-30

**Soundness:** 3
**Presentation:** 1
**Contribution:** 2
**Rating:** 2
**Confidence:** 4

**Summary:**

This paper addresses predicting diabetic complications from longitudinal patient traces extracted from administrative claims in Brazil’s TUSS billing-code ecosystem, which contain irregular events, sparsity, and noise. The authors propose MedAttention, which combines BiLSTM, Self-attention pooling, and embeddings of TUSS code events and absolute time. MedAttention was evaluated on two administrative claim datasets from different operators and performed better than baselines, with absolute time embeddings and self-attention working in complementary ways.

**Strengths:**

-- Large-scale analysis for predicting diabetic complications is quite a significant problem, and its importance is well-justified in the paper.

-- Experimental results on multiple datasets containing 3.9 million anonymized patient records demonstrated the effectiveness of the proposed method.

**Weaknesses:**

-- The proposed method may have novel points and have some practical impact, but the clarity issues are too severe to understand the method:
* In l.157, "6–12 months ahead" is vague.
* In l.159 and l.160, "outcome" and "index date" are not defined, although they look like important concepts.
* l.160, the important definition of positive sequence is vague: "Positive sequences end ≥180 days before the first complication."
* In l.196, the description of TUSS code embeddings is not sufficient and clear enough to understand.
* In l.201, "date", "t", "d", and "i" in the equation are not defined. Why P=10,000days?
* In l.204, "tuss" is not defined.
* In l.206, how are demographics appended?
* Section 3.3 is hard to follow. MedAttention should be named in the main text.
* In l.269, how is Operator 2 validation split?
* Sections 3.6 and 4.1 are duplicated mostly.

-- Paragraph started from l.130 can be moved to the Ethics statement section since it is less related to the main discussion of the paper.

-- In l.154, the proxy diabetes cohort can be subjective. It is better to justify this process with a reference.

-- In l.192, it is better to justify the oversampling by a reference or an ablation.

-- Miner issues:
* \citet{} or \citep{} should be used for citation.
* Table 1  and Figure 3 should be described in the main text.
* l.188, what is "SD 17"?
* Tables and figures are better located at the top.

**Questions:**

Please see above.

---

### Official Review · Reviewer_j8Fv · 2025-11-01

**Soundness:** 2
**Presentation:** 2
**Contribution:** 2
**Rating:** 2
**Confidence:** 4

**Summary:**

ChatGPT said:This paper introduces MedAttention, a time-aware deep learning model for forecasting severe diabetic complications from large-scale Brazilian healthcare billing (TUSS) data.
The model combines skip-gram embeddings of medical codes, sinusoidal time embeddings, and a BiLSTM with self-attention pooling to capture long, irregular patient histories.

**Strengths:**

- The paper has significant real-world meaning, as it demonstrates strong performance on a massive national claims dataset.

- Ablation study is included to support the architecture design of the model.

**Weaknesses:**

- Limited methodological novelty: The model architecture largely combines existing components (skip-gram embeddings, sinusoidal time encoding, BiLSTM with attention) rather than introducing new algorithms or learning objectives.

- The split of training and test set is not well-defined. The beginning of PR-curve in Fig 6 suggests strong over-fitting and data leakage.

**Questions:**

See weakness.

---

### Meta-Review · Area_Chair_dqx4 · 2025-12-27

**Summary:**

The paper proposes MedAttention, a deep learning framework designed to forecast severe diabetic complications (angiopathy, amputation, renal failure) using Brazilian administrative billing codes (TUSS). The model architecture combines skip-gram embeddings for medical codes, fixed sinusoidal embeddings for absolute time, and a BiLSTM backbone with self-attention pooling to handle irregular patient histories. The authors evaluate the model on a private dataset of approximately 3.9 million beneficiaries.


The paper received three negative reviews, and the authors did not submit a rebuttal. Consequently, all reviewer concerns remain outstanding. The consensus is that while the dataset scale is impressive, the work suffers from critical methodological, experimental, and presentation flaws.

**Reviewer Concerns:**

The paper presents a large-scale case study, but the methodological novelty is limited, and the experimental rigor is insufficient. The lack of standard baselines, potential data leakage, and clarity issues regarding cohort definition make it unsuitable for publication in its current state. The authors did not engage with the review process to address these significant concerns.

**Reviewer Scores:**

Because the authors did not submit a rebuttal, I don't think reviewers will change their scores.

---

### Decision · Program_Chairs · 2026-01-26

Reject